# Pruritus in Chronic Cholestatic Liver Diseases, Especially in Primary Biliary Cholangitis: A Narrative Review

**DOI:** 10.3390/ijms26051883

**Published:** 2025-02-22

**Authors:** Tatsuo Kanda, Reina Sasaki-Tanaka, Naruhiro Kimura, Hiroyuki Abe, Tomoaki Yoshida, Kazunao Hayashi, Akira Sakamaki, Takeshi Yokoo, Hiroteru Kamimura, Atsunori Tsuchiya, Kenya Kamimura, Shuji Terai

**Affiliations:** 1Division of Gastroenterology and Hepatology, Uonuma Institute of Community Medicine, Niigata University Medical and Dental Hospital, Uonuma Kikan Hospital, Minamiuonuma 949-7302, Japan; 2Division of Gastroenterology and Hepatology, Graduate School of Medical and Dental Sciences, Niigata University, Niigata 951-9510, Japankhayashi@med.niigata-u.ac.jp (K.H.); saka-a@med.niigata-u.ac.jp (A.S.);; 3Department of Gastroenterology and Hepatology, Faculty of Medicine, University of Yamanashi, Chuo 409-3898, Japan; a.tsuchiya@yamanashi.ac.jp; 4Department of General Medicine, Niigata University School of Medicine, Niigata 951-9510, Japan; kenya-k@med.niigata-u.ac.jp

**Keywords:** autotaxin, ileal bile acid transporter inhibitors, liver diseases, primary biliary cholangitis, pruritus

## Abstract

Patients with chronic cholestatic liver diseases often experience itch and struggle with this symptom. We discuss the mechanism of itch in patients with chronic cholestatic liver diseases, such as primary biliary cholangitis (PBC) and others, and their therapies, including ileal bile acid transporter (IBAT) inhibitors. In patients with PBC, there are high serum/plasma concentrations of multiple factors, including bile salts, bilirubin, endogenous opioids, lysophosphatidic acid (LPA), autotaxin, and histamine. Bile salts, bilirubin, LPA, and autotaxin affect itch mediators in the skin and sensory nerves, while the endogenous opioid balance affects mediators in the spinal cord. Itch is sensitized by both the peripheral and central nervous systems. Both mechanisms are involved in itch in patients with chronic cholestatic liver disease. Although IBAT inhibitors have been approved for use in pediatric cholestatic conditions, such as progressive familial intrahepatic cholestasis and Alagille syndrome, IBAT inhibition seems to be a promising treatment for chronic refractory itch in patients with PBC. A traditional non-systematic review results in this narrative review. Multidisciplinary cooperation, involving hepatologists, dermatologists, and pharmacists, could provide better treatment for PBC patients suffering from refractory itch. In conclusion, we summarized the existing knowledge on itch caused by chronic cholestatic liver diseases, especially in PBC with a focus on the mechanisms and therapies. This narrative review provides the mechanisms and therapeutic options for itch in patients with chronic cholestatic liver diseases.

## 1. Introduction

Chronic itch, namely pruritus, is caused by various clinical manifestations and diseases such as dermatological, hepatobiliary, renal, endocrine, metabolic, hematological, malignant, neurological, psychological, and other disorders [1]. Chronic itch can adversely affect the quality of life. It is important to avoid chronic itch when primary changes are treated.

In total, 200 (88.9%) out of 225 patients with primary biliary cholangitis (PBC) self-reported pruritus of any severity [2]. Compared with patient-reported measures, itch in PBC is often under-recorded in medical records (88/225 (39.1%) vs. 120/225 (53.3%)), and is associated with lower patients’ health-related quality of life. The reasons why itch is under-recorded are considered as follows: (1) Physicians may be unfamiliar with available guidelines for recognizing and treating pruritus; (2) It is possible that the management of PBC in clinical practice could be recorded in medical records, but associated conditions such as pruritus/itch were not; (3) It is possible that many PBC patients do not recall having their pruritus evaluated or discussing itch with their providers, etc. [2]. Therefore, it is possible that physicians may be unaware of their itch, resulting in potential under-treatment.

Chronic itch is one of the most common symptoms associated with PBC [3], affecting up to 75% of patients at some point during their disease course [4], although the severity of PBC does not always seem to correlate with symptoms [5]. Non-overweight patients with PBC tend to experience symptoms, such as itching or fatigue. Gungabissoon et al. reported that 139 (6.8%) of 1963 patients with PBC had pruritus as a symptom of interest [4]. Itch remains under-treated, highlighting a need for treatments specifically indicated for cholestatic itch. A total of 170 (81%) of 211 PBC patients who completed a patient-reported outcome (PRO) survey reported pruritus, and 70 (33%) had never received treatment for their itch [6]. Thus, the fact that itch in PBC is undertreated may be partly due to the ineffectiveness of current treatments, poor tolerance, or the lack of FDA-approved medications for itch.

Although patients with chronic non-cholestatic liver disease suffer from itch [7,8,9,10,11,12], we focus on itch in patients with chronic cholestatic liver disease. Underlying liver disease, aspartate aminotransferase levels equal to or greater than 60 U/L, and comorbid diabetes are risk factors associated with pruritus in patients with chronic liver disease.

Thus, patients with chronic cholestatic liver disease often suffer from itch. This narrative review results from a traditional and non-systematic literature review, using PubMed. This review summarizes the existing knowledge on chronic itch caused by chronic cholestatic liver diseases, especially PBC, which is an autoimmune liver disease, with a focus on the underlying mechanisms and treatment.

## 2. Mechanism of Chronic Itch in Patients with Liver Diseases

In patients with chronic liver diseases, itch is sensitized by both the peripheral and central nervous systems [13,14]. Itch is a sensation, which emanates from the skin and is transferred through peripheral nerve fibers to the central nervous system [13].

In patients with PBC, there are high serum/plasma concentrations of multiple factors, including bile salts, bilirubin, endogenous opioids, lysophosphatidic acid (LPA), autotaxin, and histamine. Bile salts, bilirubin, LPA, and autotaxin affect itch mediators in the skin and sensory nerves, while endogenous opioid balance affects mediators in the spinal cord [13].

Bile salts are able to bind to the Takeda G protein-coupled receptor 5 (TGR5; G protein-coupled bile acid receptor 19), which is expressed by neurons in the dorsal root ganglia. Bile acids can activate these neurons and sensory nerves as well [15]. Bile acids activate TGR5 on sensory nerves, stimulating the release of neuropeptides in the spinal cord that transmit itch [14]. However, this pathway seems to be completely deactivated in the rodent model of cholestasis [16]. Bile acids and bilirubin are ligands for the Mas-related G protein-coupled receptor X4 (MRGPRX4) and play a role in cholestatic itch [17]. MRGPRX4 is the only bile acid receptor expressed on the membranes of human itch neurons. MRGPRX4 is expressed in human dorsal root ganglion neurons and co-expresses with the itch H1 histamine receptor (H1R) [18] (Figure 1). Yu et al. reported that human TGR5 is not expressed in human dorsal root ganglions [18]. A recent study demonstrated that 3-sulfated bile acids are elevated in cholestatic patients with itch and the cryo-electron microscopy structure of MRGPRX4 [19]. Interaction between MRGPRX4 and transient receptor potential ankyrin 1 (TRPA1)/transient receptor potential vanilloid 1 (TRPV1) channels may be involved in chronic itch symptoms.

Obeticholic acid is a farnesoid X receptor (FXR) agonist used for the treatment of patients with PBC and results in the accumulation of bile acids. Obeticholic acid regulates the de novo synthesis of primary bile acids. Primary bile acids are synthesized from cholesterol, and FXR-dependent inhibition of de novo bile acid synthesis could lead to the accumulation of cholesterol [20]. However, pruritus, constipation, diarrhea, and hyperlipidemia were major adverse events [20]. Further studies are needed. Bilirubin elicits an itch sensation by directly stimulating peripheral nerve fibers [21]. LPA and its enzyme autotaxin are also associated with itch [22]. Hepatocyte autotaxin expression, which results in increased LPA levels, activation of hepatic stellate cells (HSCs), and amplification of profibrotic signals, promotes liver fibrosis and hepatocarcinogenesis, suggesting that autotaxin/LPA is the causative link in cirrhosis and hepatocellular carcinoma (HCC) [23,24,25]. The autotaxin/LPA pathway plays a role in pathogenesis and pruritus in chronic cholestatic liver diseases, including PBC. Serum autotaxin levels may serve as a predictive marker for liver-related events in Japanese patients with PBC [26]. LPA is an itch mediator, and dorsal root ganglion neurons directly respond to LPA depending on TRPA1/TRPV1 [27] (Figure 1).

The liver accumulates and excretes opioids, and damage to the liver elevates plasma opioid levels and μ-opioid activity [28]. The μ-opioid receptor agonist morphine induces itch, while the μ-opioid receptor antagonist naloxone inhibits morphine-induced itch and chronic cholestasis-related itch [29,30]. In contrast, the κ-opioid receptor has been shown to suppress pruritus. The κ-opioid receptor agonist nalfurafine hydrochloride suppressed pruritus induced by the intracisternal administration of morphine [29,31]. There are several itch mediators, sensitizers, and desensitizers [31,32]. Thus, opioid receptors play a role in modulating pruritus (Figure 1).

In contrast, central itch is associated with the activation of the μ-opioid receptor, while peripheral itch is induced by the activation of mast cells that release histamine, which binds to histamine receptors H1R and H4R around borders between the epidermis and dermis, subsequently activating TRPA1, TRPV1, and C fibers in the dorsal root [27,32]. Further scratching exacerbates C-fiber activation, creating a feedback loop that intensifies the sensation of itch. Thus, both the peripheral and central nervous systems are involved in the mechanisms of action of chronic itch in liver diseases (Figure 1). Although the most commonly utilized treatments for itch are antihistamines, these treatments are largely ineffective for histamine-resistant itch. Both histamine and non-histamine pathways are also important for the mechanism of itch.

## 3. Treatment for Chronic Itch in Patients with Primary Biliary Cholangitis

The first standard therapy for PBC patients is ursodeoxycholic acid (UDCA), which could lead to an improvement in liver dysfunction and symptoms such as fatigue and itching [33,34,35,36,37,38,39,40]. UDCA is especially effective for patients with an early stage of PBC. UDCA replaces hydrophobic bile acids with hydrophilic bile acids, stimulates the biliary secretion of bile acids into the biliary tract, improves enterohepatic circulation, and prevents hepatocytes from apoptosis.

UDCA has lowering effects on aminotransferase, gamma-glutamyl transpeptidase, alkaline phosphatase, total cholesterol, as well as IgM. The proportion of patients with chronic itch requiring the use of cholestyramine was significantly lower at 2 years than at baseline [41].

Among fibrates, bezafibrate combined with UDCA improved the short-term efficacy of Japanese patients with PBC [42,43,44]. Bezafibrate is a ligand of peroxisome proliferator-activated receptor alpha, which is involved in immune function. However, with the use of bezafibrate, concentrations of the principal serum bile acids did not change significantly because bezafibrate is generally used in the combination with UDCA [42]. Itching improved in some PBC patients treated with the combination of bezafibrate and UDCA for 6 months [43]. Bezafibrate improves the elevation of alkaline phosphatase levels in PBC patients with a refractory response to UDCA [44,45,46]. Adherence to medication may also be a more important contributor to the efficacy of the combination therapy of UDCA and bezafibrate.

In total, 3162 (81%)/3908 PBC patients received UDCA only, while 746 (19%)/3908 received UDCA and bezafibrate over 17,360 and 3932 patient-years, respectively. During follow-up, 183 deaths (89 liver-related) and 21 liver transplantations were registered. Exposure to combination therapy was associated with a significant decrease in all-cause and liver-related mortality or the need for liver transplantations (adjusted hazard ratios: 0.3253, 95% CI 0.1936–0.5466, and 0.2748, 95% CI 0.1336–0.5655, respectively; *p* < 0.001 for both), resulting in an improved prognosis in the combination of bezafibrate and UDCA group [46]. The addition of bezafibrate to UDCA was associated with improved prognosis for the long-term periods [47,48].

A meta-analysis demonstrated that combination therapy with UDCA and fenofibrate was more effective in reducing alkaline phosphatase than UDCA monotherapy [49,50]. Fibrates appear to be safe and well tolerated in patients with PBC, with a low frequency of adverse events [51]. Fibrates can significantly improve pruritus symptoms in a subset of patients with PBC [52].

Pemafibrate, which is safer for patients with chronic kidney disease than bezafibrate, also demonstrates efficacy in patients with PBC [53,54]. Further studies about the effect of pemafibrate on chronic itch are needed. In four cases that switched from bezafibrate to pemafibrate, alkaline phosphatase level significantly decreased (*p* = 0.031), and γ-glutamyl transferase level tended to decrease (*p* = 0.063) over the 3 months after the addition of pemafibrate [53]. The addition of pemafibrate is effective in PBC patients with dyslipidemia who are resistant to UDCA monotherapy [54]. The efficacy of the addition of pemafibrate to UDCA should be examined in PBC patients without dyslipidemia.

Obeticholic acid, an FXR agonist, has been shown to improve the elevation of alkaline phosphatase in PBC patients who have not responded well enough to UDCA [55]. Obeticholic acid is derived from the primary human chenodeoxycholic acid. Phase 3 clinical trials demonstrated that obeticholic acid, administered with ursodiol or as monotherapy for 12 months in PBC patients, resulted in decreases from baseline in alkaline phosphatase and total bilirubin levels compared with placebo groups [56]. Obeticholic acid has been approved as one of the second-line therapies for PBC patients. Although most PBC patients with pruritus can be effectively managed to minimize discontinuation of obeticholic acid, pruritus is the most common side effect, limiting treatment at higher doses [57,58].

Some PBC patients with chronic itch additionally seem to use antipruritic medication (53.3%, 120/225), bile acid sequestrants (12.9%, 29/225), sertraline (13.8%, 31/225), rifampin (rifampicin) (3.1%, 7/225), naltrexone/naloxone (5.8%, 13/225), antihistamines (42.7%, 96/225), or other medications (2.2%, 5/225) [2]. Evidence of corticosteroids or mycophenolate is limited, although some evidence of azathioprine and cyclosporine exists [59,60]. Of course, antihistamines and antibiotics are also effective as needed [34].

Antagonists of the μ-opioid receptor, such as naltrexone and naloxone, are effective for cholestasis-associated itch [61,62]. Rifampin (standardized mean difference (SMD) −1.62, 95% CI −3.05 to −0.18) and opioid antagonists (SMD −0.68, 95% CI −1.19 to −0.17) significantly reduced cholestasis-related pruritus [61]. Similarly, the use of the κ-opioid receptor agonist nalfurafine hydrochloride could play a role in the treatment of chronic severe itch [63,64]. Serum autotaxin levels did not decrease along with the improvement in pruritus from nalfurafine treatment in PBC patients [63]. Nalfurafine hydrochloride is effective for chronic itch in patients undergoing hemodialysis [64]. Nalfurafine hydrochloride should be avoided in patients with acute exacerbation or chronic hepatic failure. The combination of nalfurafine hydrochloride and antihistamine is more effective accordingly [34]. Here, multidisciplinary cooperation, involving hepatologists, dermatologists, pharmacists, and others, may also be important for seeing and treating PBC patients with refractory itch across many different fields [65,66].

## 4. Emerging Therapies

### 4.1. Ileal Bile Acid Transporter (IBAT) Inhibitors for Itch in Patients with Chronic Cholestatic Liver Diseases, Such as Primary Biliary Cholangitis and Others

#### 4.1.1. Mechanism of IBAT Inhibitors

Recently, inhibitors of the ileal bile acid transporter (IBAT: apical sodium-dependent bile acid transporter (ASBT)) have been noticed as a novel class of agents in the treatment options of chronic itch and others [67]. IBAT functions as a symporter that uses the natural osmotic sodium gradient to actively cotransport conjugated bile acids into the cell cytosol. The primary site of bile acid reabsorption occurs in the terminal ileum, where IBAT is overexpressed by nearly 50-fold [68]. IBATs are efficacious, reabsorbing roughly 95% of bile acids in the small bowel [69]. Conjugated bile acid uptake is mediated by IBAT in the ileum [68]. The Na^+^/bile acid cotransport system is a major regulator of serum cholesterol homeostasis [70].

IBAT is a bile acid sodium symporter protein that is encoded by the solute carrier (SLC) family 10 member 2 (*SLC10A2*, also known as the apical sodium-dependent bile acid transporter (*ASBT*)) gene in humans [71]. *SLC10A2* mutations can cause primary bile acid malabsorption, and the ileal Na^+^/bile acid cotransporter’s role is important in the intestinal reclamation of bile acids [72].

IBAT inhibitors target the ileal re-uptake of bile acids, prevent the enterohepatic recirculation of bile acids, and reduce the total bile acid pool size and exposure of the liver [62]. IBAT inhibitors have been approved for use in pediatric cholestatic conditions, progressive familial intrahepatic cholestasis, and Alagille syndrome [73,74,75,76]. Cholestyramine was the only US FDA-approved drug for cholestatic pruritus until the FDA approved IBAT inhibitors for use in progressive familial intrahepatic cholestasis and Alagille syndrome [73]. Both cholestyramine and IBAT inhibitors decrease the bile acid pool [73] (Figure 2).

#### 4.1.2. Therapies for Intrahepatic Cholestasis of Pregnancy

Intrahepatic cholestasis of pregnancy is one of the unique liver diseases during pregnancy [77]. The diagnosis of intrahepatic cholestasis of pregnancy is based on a serum bile acid level of more than 10 μmol/L in the presence of pruritus, in general, during the second or third trimester. Serum bile acid levels correlate with the risk of stillbirth, with the highest risk in 18 (3.44%) of 524 patients with a serum bile acid level equal to or more than 100 μmol/L (vs. 3 (0.13%) of 2310 patients with intrahepatic cholestasis of pregnancy of less than 40 μmol/L and 4 (0.28%) of 1412 patients with intrahepatic cholestasis of pregnancy of 40–99 μmol/L; *p* < 0.0001) [78].

Treatment should be offered with oral ursodeoxycholic acid in a daily divided dosage to total 10–15 mg/kg/day, to improve itch, serum bile acid levels, serum alanine aminotransferase levels, and decrease outcomes, including preterm birth and stillbirth [79]. Cholestyramine and rifampin could be used as additional treatments for itch [77].

After 35 weeks gestation, an elective early delivery should be planned at the stage of pregnancy with post-prandial serum bile acid concentrations equal to or more than 100 μmol/L, to reduce the risk of fetal death [80].

It was reported that placental mRNA expression of bile acid transporters, including *SLC10A2*, *SLCOA1*, and ATP-binding cassette (ABC) subfamily C member 2 (*ABCC2*), was positively correlated with bile acid concentrations in intrahepatic cholestasis of pregnancy [81]. Placental *SLC10A2* mRNA was also correlated with maternal body mass index. The use of IBAT inhibitors should currently be avoided because their safety during pregnancy has not yet been established. 

#### 4.1.3. IBAT Inhibitors for Children with Chronic Cholestatic Liver Diseases

In Alagille syndrome and progressive familial intrahepatic cholestasis, which are rare inherited cholestatic liver diseases, disruption of secretion of bile acids results in the accumulation in the liver, leading to underlined pruritus and exacerbated liver damage [76]. One method to decrease pathological bile acid accumulation in the body is surgical biliary diversion, which interrupts the enterohepatic circulation, using diverting bile acids to an external stoma. Another new method is the non-surgical inhibition of IBAT by IBAT inhibitors. These methods could normalize serum bile acids and reduce itch and inflammation of the liver, resulting in the improvement of quality of life in patients with Alagille syndrome or progressive familial intrahepatic cholestasis [76].

In children with Alagille syndrome, maralixibat, an apical, sodium-dependent, and bile acid transport inhibitor, is the first IBAT inhibitor that has shown durable and clinically meaningful improvements in cholestasis [74]. Significant improvements in pruritus were seen with maralixibat at week 48 of the ICONIC study, which is a phase 2 study featuring a 4-week double-blind, placebo-controlled, and randomized drug withdrawal period in children with Alagille syndrome experiencing moderate-to-severe pruritus, and these are associated with an improved health-related quality of life [82]. Odevixibat could be an efficacious non-surgical intervention to improve pruritus, reduce serum bile acids, and enhance the standard of care in patients with Alagille syndrome [83].

Positive phase 3 results for odevixibat were reported for progressive familial intrahepatic cholestasis [75]. Odevixibat was well tolerated, and the most common treatment-emergent adverse event was diarrhea or frequent bowel movements [84]. In children with progressive familial intrahepatic cholestasis, odevixibat effectively reduced pruritus and serum bile acids [85]. Bile acid synthesis in the liver, enterohepatic circulation of bile acids, and the mode of action of IBAT inhibitors are illustrated in ref [84].

#### 4.1.4. IBAT Inhibitors for Adults with Chronic Cholestatic Liver Diseases, Such as PBC

The change in total serum bile acid (TSBA), according to the area under the TSBA concentration curve over 24 h (AUC0-24), correlates significantly with and can be predictive of pruritus improvement in PBC patients treated with linerixibat [86]. In patients with PBC, chronic itch is associated with an elevation in serum bile acid and autotaxin, which decrease after modification via IBAT inhibition. In PBC patients with pruritus, a 14-day treatment with IBAT inhibitor GSK2330672 demonstrated efficacy in reducing the pruritus severity without serious adverse events [87]. Linerixibat had a non-significant effect on itch in the primary intent-to-treat analysis, but this was associated with a significant dose-dependent reduction in itch in the per-protocol population [88]. Primary sclerosing cholangitis (PSC) is also frequently associated with pruritus. IBAT inhibitors for itch in PSC patients seem useful and improve their quality of life [89].

The recent development of IBAT inhibitors for patients with PBC and chronic itch sheds new light on the treatment of chronic itch in these patients [90,91,92,93,94]. The 2-week administration of GSK2330672, an IBAT inhibitor, had an effect on the fecal microbiomes in PBC patients. GSK2330672 increased the relative abundance of Firmicutes (*p* = 0.033) and Clostridia (*p* = 0.04) and reduced Bacteroidetes (*p* = 0.033) and Bacteroidia (*p* = 0.04) [92]. As the gut–liver axis plays a role in the homeostasis of the liver, the gut microbiome interplays with a diverse spectrum of hepatic changes, including steatosis, inflammation, fibrosis, cholestasis (including itch), and tumorigenesis [95].

IBAT inhibitors are used for the treatment of chronic constipation and cholestatic pruritus caused by PBC and NASH [86,96]. Nonalcoholic fatty liver disease and steatohepatitis are associated with gut microbiota [97,98]. Treatment with an IBAT inhibitor significantly improved hepatic steatosis in high-fat diet mice, and fecal microbiome transplantation using stool from high-fat diet plus IBAT inhibitor mice prevented hepatic steatosis caused by a high-fat diet [99,100]. Table 1 shows the IBAT inhibitors for various diseases. Of interest, common adverse events are diarrhea and abdominal pain. As constipation exacerbates hepatic encephalopathy, it may be useful for patients with PBC and hepatic encephalopathy to use IBAT inhibitors. Further studies will be needed at this point.

Representative clinical trials and their comparative efficacy are shown in Table 1.

### 4.2. Ileal Stem Cell Transplantation

Ileal stem cell transplantation may also be useful for itch of chronic cholestatic liver diseases [101,102]. The area of peak IBAT function was found to be located in the terminal ileum in rodents [102]. Ileal stem cell clusters were used to establish a new zone of bile acid uptake and IBAT expression in a jejunal segment in adult Lewis rats [101].

## 5. Other Treatments for Itch in Chronic Cholestatic Liver Diseases

Ultraviolet B (UVB) phototherapy appears to be a promising and well-tolerated treatment for cholestasis-associated pruritus or PBC-associated pruritus [103,104]. Liver transplantation for pruritus is highly effective, but fatigue does not disappear in the majority of the patients, although these two common symptoms are generally observed in patients with PBC and liver transplantation [105]. Patients transplanted for PBC suffer more frequently from acute and late cellular rejections. Long-term administration of UDCA following liver transplantation has a beneficial effect on the recurrence of PBC [105].

Biological therapies were also performed [106], and pruritus was improved in 60% of patients at 12 months [107]. Depletion of B cells with the anti-CD20 monoclonal antibody rituximab influences the induction, maintenance, and activation of both B and T cells, providing a potential mechanism for the treatment of patients with PBC and an incomplete response to UDCA [106]. Selective B-cell depletion with rituximab was safe and associated with a significant decrease in antimitochondrial antibody production [107].

Cholestyramine, one of the anion-exchange resins, brings improvement in the itch in PBC patients [108]. Cholestyramine sequesters bile acids in a resin complex for excretion to decrease bile acid reabsorption in the distal small bowel [109]. Bile acid sequestrant cholestyramine eliminates pruritogens, resulting in the improvement of itch [110]. As a result of decreased reuptake of bile acids in the distal small bowel, cholestyramine decreased the accumulation of bile acids [109]. Thus, despite its ability to alleviate itch in some patients, complete resolution of chronic itch is rare. Adverse events are malabsorption of cholestyramine fats and fat-soluble vitamins [109].

Rifampin is occasionally used for the treatment of cholestatic itch and improves chronic itch in some patients with chronic liver diseases [109]. Rifampin is also useful for short-term relief of itch in PBC patients [111]. Rifampin significantly reduced itch intensity and autotaxin activity, which are specific to itch caused bycholestasis, in patients with itch not responding to bile salt sequestrants [112]. Rifampin also inhibited autotaxin expression in human hepatoblastoma HepG2 cells and hepatoma cells overexpressing the pregnane X receptor (PXR) [112]. The beneficial antipruritic action of rifampin partly depends on the PXR-dependent transcriptional inhibition of autotaxin expression.

Representative treatments for patients with PBC and itch are shown in Table 2.

The diagnosis and treatment of PBC have developed, and the prognosis for PBC has recently improved, resulting from the earlier diagnosis [113,114]. Recently, patients with PBC have been characterized by older age at diagnosis, an increase in male-to-female ratio, higher response rates of UDCA, and longer survival, resulting from the early recognition of this disease [113]. A machine learning prediction model for treatment responders in PBC patients with refractory itch may be useful and provide new strategies [115,116]. Recently, the molecular mechanism of enterohepatic recirculation of bile acids has been elucidated; further studies are needed regarding this point [117,118]. Elafibranor and seladelpar are peroxisome proliferator-activated receptor agonists recently approved for use in patients with PBC [119,120,121]. Seladelpar decreased serum interleukin-31 (IL-31), which is a cytokine known to mediate pruritus, and bile acids, resulting in the improvement of pruritus in PBC patients [122]. Further studies on itch in chronic cholestatic liver disease are currently in progress [90,121,123,124,125].

## 6. Future Perspectives

In patients with PBC, there are high serum/plasma concentrations of multiple factors, including bile salts, bilirubin, endogenous opioids, LPA, autotaxin, and histamine. Bile salts, bilirubin, LPA, and autotaxin affect itch mediators in the skin and sensory nerves, and the endogenous opioid balance affects mediators in the spinal cord. Itch is sensitized by both the peripheral and central nervous systems. Both mechanisms are involved in chronic itch in patients with liver diseases. IBAT inhibition seems to be a promising treatment for chronic refractory itch in patients with PBC.

## 7. Conclusions

This review summarized the existing knowledge on itch caused by chronic cholestatic liver diseases, especially PBC, with a focus on the underlying mechanism and treatment. This narrative review provided the mechanism and therapeutic options for itch in patients with chronic cholestatic liver disease, who struggle with this symptom. Multidisciplinary cooperation, involving hepatologists, dermatologists, and pharmacists, is important for seeing and treating PBC patients with refractory itch.

## Figures and Tables

**Figure 1 ijms-26-01883-f001:**
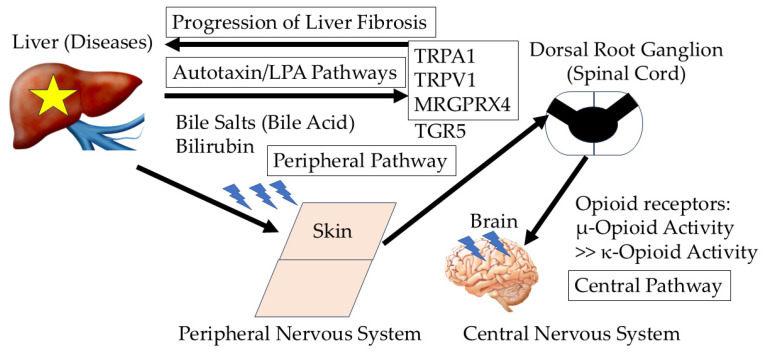
Representative mechanism of itch in chronic cholestatic liver diseases. Both the peripheral and central nervous systems are involved in chronic itch from liver diseases. LPA, lysophosphatidic acid; TRPA1, transient receptor potential ankyrin 1; TRPV1, transient receptor potential vanilloid 1; MRGPRX4, Mas-related G protein-coupled receptor X4; TGR5, Takeda G protein-coupled receptor 5 (G protein-coupled bile acid receptor 19). Yellow star, diseases; Blue lightning, itch.

**Figure 2 ijms-26-01883-f002:**
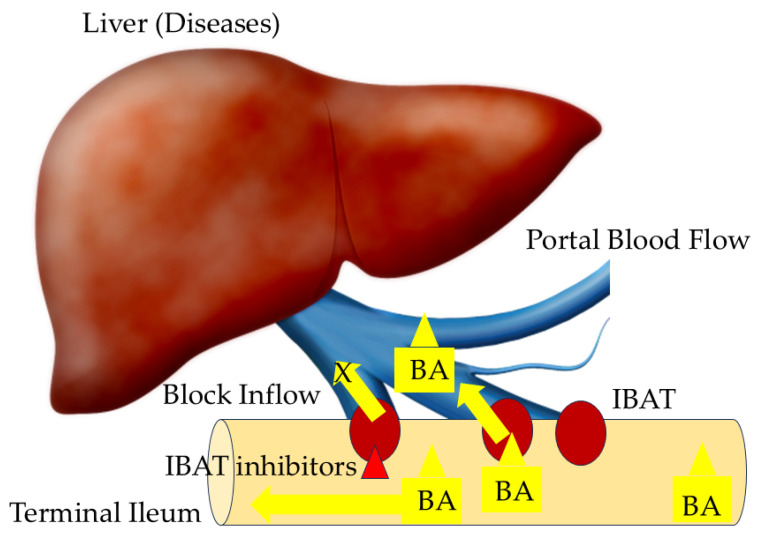
Mechanism of ileal bile acid transporter (IBAT) inhibitors. IBAT is involved in the absorption of bile acid (BA) from the terminal ileum to the portal blood flow. IBAT inhibitor blocks the inflow of BA into the portal vein. Red triangle, IBAT inhibitors; Brown circles, IBAT.

**Table 1 ijms-26-01883-t001:** Representative ileal bile acid transporter inhibitors for various diseases or their itch.

Drugs	Diseases	Critical Metrics	Common Adverse Events (AEs)	References
Maralixibat	Alagille syndrome	From baseline to week 48, serum bile acid (−96 μmol/L, −162 to −31) and pruritus (−1.6 pts, −2.1 to −1.1) improved (phase 2 study)	Diarrhea, abdominal pain	[74]
Odevixibat	Progressive familial intrahepatic cholestasis	Odevixibat significantly reduced pruritus and serum bile acid levels (phase 3 study)	Diarrhea, frequent bowel movements	[75]
Maralixibat	Alagille syndrome	Significant improvements in pruritus, associated with improved health-related quality of life (phase 2 study)	N/A	[76,82]
Odevixibat	Alagille syndrome	Odevixibat resulted in significantly greater reductions in mean serum bile acids from baseline, which was associated with mean scratching scores at weeks 21–24 (phase 3 study)	Diarrhea (29%), pyrexia (23%)	[83]
Odevixibat	Progressive familial intrahepatic cholestasis	Odevixibat reduced pruritus and serum bile acids vs. placebo (phase 3 study)	Diarrhea, frequent bowel movements, fever	[85]
Linerixibat (GSK2330672)	Primary biliary cholangitis	GSK2330672 produced significantly greater reduction from baseline in the 0 to 10 numerical rating scale (NRS) (−23%, *p* = 0.037), Primary biliary cholangitis −40 itch domain, (−14%, *p* = 0.034), and 5-D itch scale (−20%, *p* = 0.0045) vs. placebo. (phase 2a study)	Diarrhea	[87]
Linerixibat	Primary biliary cholangitis	Linerixibat effect on itch was associated with a significant dose-dependent reduction in itch (phase 2b study)	Diarrhea	[88]
A4250	Primary biliary cholangitis	Remarkable improvement in pruritus	Abdominal pain, diarrhea	[91]
Maralixibat	Primary sclerosing cholangitis	Maralixibat was associated with reduced serum bile acid levels, which are associated with significant improvement in pruritus	Diarrhea	[89]
Volixibat (SHP626)	Nonalcoholic steatohepatitis	SHP626 increased mean total fecal BA excretion about ~1.6–3.2 times in healthy volunteers and ~8 times in patients with T2DM vs. placebo.	Mild or moderate gastrointestinal adverse events	[84]

N/A, not available.

**Table 2 ijms-26-01883-t002:** Treatments for patients with primary biliary cholangitis and itch.

Drugs	References
Ursodeoxycholic acid	[33,34,35,36,37,38,39,40,41]
Bezafibrate	[42,43,44,45,46,47,48]
Fenofibrate	[49,50,51,52]
Pemafibrate	[53,54]
Obeticholic acid	[55,56,57,58]
Cholestyramine	[108,109,110]
Rifampin	[111,112]
Naltrexone/Naloxone	[61,62]
Nalfurafine hydrochloride	[34,62,63]
IBAT inhibitors	[87,88,89,91]
Ultraviolet B phototherapy	[103,104]
Liver transplantation	[105]

IBAT, ileal bile acid transporter.

## Data Availability

Not applicable.

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
