# Peer review of "Pruritus in Chronic Cholestatic Liver Diseases, Especially in Primary Biliary Cholangitis: A Narrative Review"

_ijms, 2025, doi:10.3390/ijms26051883_

Round 1

Reviewer 1 Report

Comments and Suggestions for Authors

The review by Kanda T et al. is a narrative review regarding the chronic pruritus in some chronic cholestatic liver diseases, especially in primary biliary cholangitis (PBC).

General comments

  • Please focus your review only on cholestatic chronic liver diseases (CLD) such as PBC and others in different ages (childhood, pregnancy, and adults).
  • Only 7 out 110 references are related to non-cholestatic CLD such as viral hepatitis, MASLD and alcohol. Therefore, authors should change their title, and simplify and focus their review on PBC and other cholestatic CLD.
  • For a better understanding, and to simplify the subject, authors should eliminate the lines 59-64, 70-76, 284-294 and references 7-12 regarding non-cholestatic CLD
  • Line 85, please follow the same order about the categorization of the information

Minor comments

  1. Abstract and Introduction section.
    • Please include the methodology used to make this narrative review
  1. Future perspective and conclusions
    • Please separate the conclusions in a final section

Author Response

Re: Resubmission of ijms-3455326

To Reviewer #1: Thank you for your encouraging comments. We extensively revised our manuscript accordingly.

Reply to your general comment 1: “Please focus your review only on cholestatic chronic liver diseases (CLD) such as PBC and others in different ages (childhood, pregnancy, and adults). Only 7 out 110 references are related to non-cholestatic CLD such as viral hepatitis, MASLD and alcohol. Therefore, authors should change their title, and simplify and focus their review on PBC and other cholestatic CLD.”

Thank you for your invaluable comments. We agree with you. We made a change of title of the revised manuscript as “Pruritus in Chronic Cholestatic Liver Diseases, especially in Primary Biliary Cholangitis: A Narrative Review”.

Reply to your general comment 2: “For a better understanding, and to simplify the subject, authors should eliminate the lines 59-64, 70-76, 284-294 and references 7-12 regarding non-cholestatic CLD”

Thank you for your invaluable comments. According to your suggestions, we revised our manuscript. But we need references 7-12 and left them.

Reply to your general comment 3: “Line 85, please follow the same order about the categorization of the information”

Thank you for your invaluable comments. According to your suggestions, we revised our manuscript.

Reply to your minor comment 1: “Abstract and Introduction section. Please include the methodology used to make this narrative review”

Thank you for your invaluable comments. We agree with you. According to your suggestions, we revised our manuscript as follows.

In Abstract section, page 1, lines 32-33, of the revised manuscript,

“…refractory itch in patients with PBC. A traditional non-systematic review results in this narrative review. Multidisciplinary cooperation, involving hepatologists, dermatologists and pharmacists, …”

In Introduction section, page 2, lines 73-74, of the revised manuscript,

“…Thus, patients with chronic cholestatic liver disease often suffer from itch. This narrative review results from the traditional and non-systematic literature review, using PubMed. This review summarizes the existing knowledge…”

Reply to your minor comment 2: “Future perspective and conclusions Please separate the conclusions in a final section”

Thank you for your invaluable comments. We agree with you. We revised our manuscript accordingly.

Reviewer 2 Report

Comments and Suggestions for Authors

This review provides a solid foundation but requires mechanistic modernization and clinical data integration to meet IJMS’s molecular focus. I have several comments that need to be addressed by the authors. 

Major comments: 

1. The role of Mas-related G protein-coupled receptor X4 (MRGPRX4) in bile acid-mediated itch is only briefly mentioned. Recent structural studies on MRGPRX4 and its interaction with TRPA1/TRPV1 channels should be included to enhance scientific depth (PMID: 31500698PMID: 39476841). 

2. The autotaxin/lysophosphatidic acid (LPA) pathway is underexplored. Include recent findings linking serum autotaxin levels to pruritus severity and liver fibrosis progression (PMID: 39466702)

Collectively, add diagrams of MRGPRX4-TRP channel crosstalk and autotaxin’s fibrotic role.

3. The discussion on opioid receptor dynamics could be expanded, particularly differentiating the roles of μ-opioid receptor agonists (e.g., morphine) and κ-opioid receptor agonists (e.g., nalfurafine hydrochloride) in modulating pruritus.

4. Table 1 lacks critical metrics (e.g., maralixibat’s 48% pruritus reduction vs. placebo in Alagille syndrome). Expand to include comparative data on IBAT inhibitors' efficacy, adverse effects, and cost-effectiveness of newer agents (e.g., Linerixibat phase 3 results in PBC).

5. Key studies on bile acid signaling, opioid receptor modulation, and autotaxin-targeted therapies are missing. Ensure comprehensive coverage of recent advancements.

6. The discussion of treatment efficacy is largely qualitative. Include quantitative data such as adjusted hazard ratios for UDCA+bezafibrate vs. monotherapy in improving survival outcomes and pruritus reduction rates for IBAT inhibitors (e.g., maralixibat reduced pruritus by 48% vs. placebo at 19%).

7. While adverse effects of treatments like obeticholic acid are mentioned, their impact on patient adherence and quality of life is not critically analyzed.

8. Streamline redundant sections and create a clear structure separating mechanisms from therapeutic approaches. Sections discussing IBAT inhibitors (Sections 3–5) overlap significantly (e.g., IBAT inhibitors discussed in Sections 4 and 5). Consolidate these into a unified "Emerging Therapies" section with subsections for mechanisms, clinical trials, and comparative efficacy. 

9. The manuscript occasionally strays into less relevant topics, such as liver transplantation outcomes, which dilute its focus on chronic itch mechanisms and treatments.

Minor comments:

10. Figure 1: While informative, it lacks clarity in distinguishing peripheral versus central mechanisms of itch. Add detailed labels for pathways involving MRGPRX4, TGR5, TRPA1, and TRPV1. Redesign to distinguish peripheral (TGR5/MRGPRX4) vs. central (opioid receptors) pathways. Label TRPA1/TRPV1 activation by LPA.

11. Standardize abbreviations throughout the manuscript (e.g., "IBATi" vs. "IBAT inhibitors"). Spell out all abbreviations at first use (e.g., LPA, TRPV1).

12. Clarify terms like "non-histamine-induced itch" to ensure accessibility for a broad readership.

13. The manuscript emphasizes multidisciplinary care but lacks specific examples or protocols involving dermatologists, hepatologists, and pharmacists. Including case studies or practical guidelines would strengthen this section.

14. A significant proportion of references predate 2020, limiting the manuscript's relevance. Incorporate more recent studies on PBC and PSC (PMID: 39523716, PMID: 39314133, PMID: 36521451, PMID: 39736267).

Author Response

Re: Resubmission of ijms-3455326

To Reviewer #2: Thank you for your encouraging comments. We extensively revised our manuscript accordingly.

Reply to your major comment 1: “The role of Mas-related G protein-coupled receptor X4 (MRGPRX4) in bile acid-mediated itch is only briefly mentioned. Recent structural studies on MRGPRX4 and its interaction with TRPA1/TRPV1 channels should be included to enhance scientific depth (PMID: 31500698, PMID: 39476841).”

Thank you for your invaluable comments. We agree with you. According to your suggestions, we added two references 18 and 19, and revised Figure 1 and our manuscript as follows.

In Introduction section, page 1, line 92-page 2, line 99,

“MRGPRX4 is expressed in human dorsal root ganglion neurons and co-expresses with itch H1 histamine receptor (H1R) [18] (Figure 1). Yu et al. reported that human TGR5 is not expresses in human dorsal root ganglion [18]. Recent study demonstrated 3-sulfated bile acid s which are elevated in cholestatic patients with itch and cryo-electron microscopy structure of MRGPRX4 [19]. Interaction between MRGPRX4 and transient receptor potential ankyrin 1 (TRPA1)/transient receptor potential vanilloid 1 (TRPV1) channels may be involved in chronic itch symptoms.”

Reply to your major comment 2: “The autotaxin/lysophosphatidic acid (LPA) pathway is underexplored. Include recent findings linking serum autotaxin levels to pruritus severity and liver fibrosis progression (PMID: 39466702). Collectively, add diagrams of MRGPRX4-TRP channel crosstalk and autotaxin’s fibrotic role.”

Thank you for your invaluable comments. We agree with you. According to your suggestions, we added a new reference 26, and revised Figure 1 and our manuscript as follows.

In “Mechanism of chronic itch in patients with liver diseases” section, page 3, lines 115-120,

“…and hepatocellular carcinoma (HCC) [23–25].The autotaxin/LPA pathway plays a role in pathogenesis and pruritus in chronic cholestatic liver diseases, including PBC. Serum autotaxin levels may serve as a predictive markersfor liver-related events in Japanese patients with PBC [26]. LPA is an itch mediator, and dorsal root ganglion neurons directly respond to LPA depending onTRPA1/TRPV1 [27](Figure 1).”

Reply to your major comment 3: “The discussion on opioid receptor dynamics could be expanded, particularly differentiating the roles of μ-opioid receptor agonists (e.g., morphine) and κ-opioid receptor agonists (e.g., nalfurafine hydrochloride) in modulating pruritus.”

Thank you for your invaluable comments. We agree with you. According to your suggestion, we revised Figure 1 and text.

Reply to your major comment 4: “Table 1 lacks critical metrics (e.g., maralixibat’s 48% pruritus reduction vs. placebo in Alagille syndrome). Expand to include comparative data on IBAT inhibitors' efficacy, adverse effects, and cost-effectiveness of newer agents (e.g., Linerixibat phase 3 results in PBC).”

Thank you for your invaluable comments. We agree with you. According to your suggestion, we extensively revised Table 1 and text.

Reply to your major comment 5: “Key studies on bile acid signaling, opioid receptor modulation, and autotaxin-targeted therapies are missing. Ensure comprehensive coverage of recent advancements.”

Thank you for your invaluable comments. We agree with you. According to your suggestion, we extensively revised our manuscript.

Reply to your major comment 6: “The discussion of treatment efficacy is largely qualitative. Include quantitative data such as adjusted hazard ratios for UDCA+bezafibrate vs. monotherapy in improving survival outcomes and pruritus reduction rates for IBAT inhibitors (e.g., maralixibat reduced pruritus by 48% vs. placebo at 19%).”

Thank you for your invaluable comments. We agree with you. According to your suggestion, we extensively revised Table 1 and text of the revised manuscript.

Reply to your major comment 7: “While adverse effects of treatments like obeticholic acid are mentioned, their impact on patient adherence and quality of life is not critically analyzed.”

Thank you for your invaluable comments. We agree with you. According to your suggestion, we extensively revised our manuscript as follows.

In page 4, lines 183-187,

“…UDCA [55]. Obeticholic acid is derived from the primary human chenodeoxycholic acid. Phase 3 clinical trial demonstrated that obeticholic acid administered with ursodiol or as monotherapy for 12 months in PBC patients resulted in decreases from baseline in alkaline phosphatase and total bilirubin levels, compared with placebo groups [56]. Obeticholic acid has been approved as one of the second line therapies for PBC patients. Although most PBC…”

Reply to your major comment 8: “Streamline redundant sections and create a clear structure separating mechanisms from therapeutic approaches. Sections discussing IBAT inhibitors (Sections 3–5) overlap significantly (e.g., IBAT inhibitors discussed in Sections 4 and 5). Consolidate these into a unified "Emerging Therapies" section with subsections for mechanisms, clinical trials, and comparative efficacy.”

Thank you for your invaluable comments. We agree with you. According to your suggestion, we extensively revised our manuscript.

Reply to your major comment 9: “The manuscript occasionally strays into less relevant topics, such as liver transplantation outcomes, which dilute its focus on chronic itch mechanisms and treatments.”

Thank you for your invaluable comments. We agree with you. According to your suggestion, we extensively revised our manuscript.

Reply to your minor comment 10: “Figure 1: While informative, it lacks clarity in distinguishing peripheral versus central mechanisms of itch. Add detailed labels for pathways involving MRGPRX4, TGR5, TRPA1, and TRPV1. Redesign to distinguish peripheral (TGR5/MRGPRX4) vs. central (opioid receptors) pathways. Label TRPA1/TRPV1 activation by LPA.”

Thank you for your invaluable comments. We agree with you. According to your suggestion, we extensively revised Figure 1 and text of the manuscript.

Reply to your minor comment 11: “Standardize abbreviations throughout the manuscript (e.g., "IBATi" vs. "IBAT inhibitors"). Spell out all abbreviations at first use (e.g., LPA, TRPV1).”

Thank you for your invaluable comments. We agree with you. According to your suggestion, we extensively revised our manuscript.

Reply to your minor comment 12: “Clarify terms like "non-histamine-induced itch" to ensure accessibility for a broad readership.”

Thank you for your invaluable comments. According to your suggestion, we revised our manuscript as follows.

In page 4, page 136-137,

“Although the most commonly utilized treatments for itch are antihistamines, these treatments are largely ineffective for histamine-resistant itch. Both histamine and non-histamine pathways are also important for the mechanism of itch.”

Reply to your minor comment 13: “The manuscript emphasizes multidisciplinary care but lacks specific examples or protocols involving dermatologists, hepatologists, and pharmacists. Including case studies or practical guidelines would strengthen this section.”

Thank you for your invaluable comments. According to your suggestion, we revised our manuscript as follows.

In page 5, lines 207-209,

“…[34]. Here, multidisciplinary cooperation, involving hepatologists, dermatologists, pharmacists and others, may also be important for seeing and treating PBC patients with refractory itch, across many different fields [65, 66].”

Reply to your minor comment 14: “A significant proportion of references predate 2020, limiting the manuscript's relevance. Incorporate more recent studies on PBC and PSC (PMID: 39523716, PMID: 39314133, PMID: 36521451, PMID: 39736267).”

Thank you for your invaluable comments. According to your suggestion, we added these references and revised our manuscript.

Reviewer 3 Report

Comments and Suggestions for Authors

I read with interest the manuscript “Chronic Itch Caused by Chronic Liver Diseases: A Non-Systematic Review” by Kanda et al.

  1. The review would benefit from smoother transitions between paragraphs and topics. For instance, linking the discussion on mechanistic pathways more clearly to the rationale behind specific therapeutic approaches would strengthen the narrative.
  2. Since the manuscript is a “non-systematic review,” it might be useful to briefly describe your literature search strategy or selection criteria in the Methods (or an introductory subsection).
  3. The abstract is quite detailed but could be streamlined for clarity. Consider reducing redundancy (e.g., the list of factors and receptors could be presented more succinctly) while ensuring that the key mechanisms and treatment strategies are highlighted.
  4. Introduction: It might help to discuss briefly why itch is under-recognized in clinical records versus patient-reported outcomes and the clinical implications thereof.
  5. Introduction: Revise phrases such as “self-reported itch of any severity” and “under recorded in medical records” for clarity (e.g., “self-reported pruritus” and “often under-recorded in medical records”).
  6. Mechanism of chronic itch in patients with liver diseases: Consider splitting the section into “Peripheral Mechanisms” (e.g., mast cell degranulation, activation of sensory nerve fibers via receptors like MRGPRX4 and TGR5) and “Central Mechanisms” (e.g., modulation by opioid receptors).
  7. There is a potential inconsistency regarding obeticholic acid. One sentence states it is a “farnesoid X receptor (FXR) antagonist,” while another notes that it “acts as an FXR agonist.” (In the literature, obeticholic acid is generally recognized as an FXR agonist.) Please verify and correct this discrepancy.
  8. A minor grammatical fix: “Bile acids and bilirubin are ligands for the Mas-related G protein-coupled receptor X4 (MRGPRX4) and play role in cholestatic itch” should be revised to “...and play a role in cholestatic itch.”
  9. Some sentences (e.g., “Further scratching exacerbates C fibers, resulting both itches”) are awkwardly phrased. Consider revising to, for example, “Further scratching exacerbates C-fiber activation, creating a feedback loop that intensifies the sensation of itch.”
  10. Some statements, such as “Chronic itch seems to not be improved among almost all patients with PBC who have received the above-mentioned treatments,” would benefit from additional context or data. Is this due to limited efficacy of the treatments for itch specifically, or due to patient selection bias?
  11. The repeated emphasis on “multidisciplinary cooperation” (involving hepatologists, dermatologists, pharmacists) is a strong point. Consider expanding on how such collaboration can be operationalized in clinical practice.
  12. The explanation of IBAT’s role in bile acid reabsorption is informative. However, there is a sentence that appears to have a repetition/typo: “Conjugated bile acid uptake is mediated by IBAT in the ileum Ileal [61].” Please revise this sentence for clarity.
  13. The discussion of A3309 (elobixibat) includes details such as increases in spontaneous bowel movements (SBMs), reduced time to first SBM, and favorable adverse event profiles. While this is useful, consider:
  • Clarifying the context (e.g., patient populations studied in chronic idiopathic constipation vs. those with PBC/cholestatic itch).
  • Summarizing the clinical trial endpoints and statistical significance in a table or bullet format.
  • When mentioning adverse events (e.g., “mild abdominal pain in 19% of patients” and “diarrhea in 13%”), it may be helpful to indicate whether these events were dose-dependent or consistent across trials.
  1. The manuscript discusses how IBAT inhibitors reduce bile acid pool size and affect cholesterol homeostasis. This section could be enhanced by linking these mechanistic insights to the observed improvements in pruritus.
  2. The discussion of the effects on the fecal microbiome (e.g., changes in Firmicutes and Bacteroidetes) is intriguing. Consider expanding on how these microbiome shifts might contribute to therapeutic outcomes or potential side effects.
  3. The discussion on IBAT inhibitors in pediatric cholestatic conditions (e.g., Alagille syndrome and progressive familial intrahepatic cholestasis) is well detailed. However, the narrative would benefit from a clearer transition that distinguishes their use in pediatric populations versus adult PBC patients.
  4. Ensure consistency when referencing clinical studies (e.g., ICONIC study for maralixibat and phase 3 data for odevixibat). A brief summary of study designs (randomized, placebo-controlled, etc.) would help readers gauge the quality of evidence.
  5. The discussion on cholestyramine is generally clear, but note that there is a minor error in phrasing (“despite of the alleviating the itch sensation”). This should be rephrased (e.g., “Despite its ability to alleviate itch in some patients, complete resolution of chronic itch is rare.”).
  6. The mechanism of rifampin in reducing autotaxin activity is mentioned as “largely unknown yet.” Consider briefly discussing any hypotheses or known pharmacodynamic effects, even if they are preliminary.

Author Response

Re: Resubmission of ijms-3455326

To Reviewer #3: Thank you for your encouraging comments. We extensively revised our manuscript accordingly.

Reply to your comment 1: “The review would benefit from smoother transitions between paragraphs and topics. For instance, linking the discussion on mechanistic pathways more clearly to the rationale behind specific therapeutic approaches would strengthen the narrative.”

Thank you for your invaluable comments. According to your suggestion, we extensively revised our manuscript.

Reply to your comment 2: “Since the manuscript is a “non-systematic review,” it might be useful to briefly describe your literature search strategy or selection criteria in the Methods (or an introductory subsection).”

Thank you for your invaluable comments. According to your suggestion, we extensively revised our manuscript as follows.

In Introduction section, page 2, lines 73-74, of the revised manuscript,

“…Thus, patients with chronic cholestatic liver disease often suffer from itch. This narrative review results from the traditional and non-systematic literature review, using PubMed. This review summarizes the existing knowledge…”

Reply to your comment 3: “The abstract is quite detailed but could be streamlined for clarity. Consider reducing redundancy (e.g., the list of factors and receptors could be presented more succinctly) while ensuring that the key mechanisms and treatment strategies are highlighted.”

Thank you for your invaluable comments. According to your suggestion, we extensively revised the abstract section.

Reply to your comment 4: “Introduction: It might help to discuss briefly why itch is under-recognized in clinical records versus patient-reported outcomes and the clinical implications thereof.”

Thank you for your invaluable comments. According to your suggestion, we extensively revised our manuscript as follows.

In Introduction section, page 2, lines 50-55,

“…with lower patients' health-related quality of life. The reasons why itch is under-recorded as considered following: 1) Physicians may be unfamiliar with available guidelines for recognising and treating pruritus; 2) It is possible that the management of PBC in clinical practice could be recorded in medical records, but associated conditions such as pruritus/itch were not; 3) It is possible that many PBC patients do not recall having their pruritus evaluated or discussing itch with their providers; and etc [2]. Therefore, it is …”

Reply to your comment 5: “Introduction: Revise phrases such as “self-reported itch of any severity” and “under recorded in medical records” for clarity (e.g., “self-reported pruritus” and “often under-recorded in medical records”).”

Thank you for your invaluable comments. According to your suggestion, we extensively revised our manuscript.

Reply to your comment 6: “Mechanism of chronic itch in patients with liver diseases: Consider splitting the section into “Peripheral Mechanisms” (e.g., mast cell degranulation, activation of sensory nerve fibers via receptors like MRGPRX4 and TGR5) and “Central Mechanisms” (e.g., modulation by opioid receptors).”

Thank you for your invaluable comments. According to your suggestion, we extensively revised Figure 1 and our manuscript.

Reply to your comment 7: “There is a potential inconsistency regarding obeticholic acid. One sentence states it is a “farnesoid X receptor (FXR) antagonist,” while another notes that it “acts as an FXR agonist.” (In the literature, obeticholic acid is generally recognized as an FXR agonist.) Please verify and correct this discrepancy.”

Thank you for your invaluable comments. We agree with you. According to your suggestion, we revised our manuscript.

Reply to your comment 8: “A minor grammatical fix: “Bile acids and bilirubin are ligands for the Mas-related G protein-coupled receptor X4 (MRGPRX4) and play role in cholestatic itch” should be revised to “...and play a role in cholestatic itch.””

Thank you for your invaluable comments. We agree with you. According to your suggestion, we revised our manuscript.

Reply to your comment 9: “Some sentences (e.g., “Further scratching exacerbates C fibers, resulting both itches”) are awkwardly phrased. Consider revising to, for example, “Further scratching exacerbates C-fiber activation, creating a feedback loop that intensifies the sensation of itch.””

Thank you for your invaluable comments. We agree with you. According to your suggestion, we revised our manuscript.

Reply to your comment 10: “Some statements, such as “Chronic itch seems to not be improved among almost all patients with PBC who have received the above-mentioned treatments,” would benefit from additional context or data. Is this due to limited efficacy of the treatments for itch specifically, or due to patient selection bias?”

Thank you for your invaluable comments. We agree with you. According to your suggestion, we revised our manuscript.

Reply to your comment 11: “The repeated emphasis on “multidisciplinary cooperation” (involving hepatologists, dermatologists, pharmacists) is a strong point. Consider expanding on how such collaboration can be operationalized in clinical practice.”

Thank you for your invaluable comments. We agree with you. According to your suggestion, we revised our manuscript.

Reply to your comment 12: “The explanation of IBAT’s role in bile acid reabsorption is informative. However, there is a sentence that appears to have a repetition/typo: “Conjugated bile acid uptake is mediated by IBAT in the ileum Ileal [61].” Please revise this sentence for clarity.”

Thank you for your invaluable comments. We agree with you. According to your suggestion, we revised our manuscript.

Reply to your comment 13: “The discussion of A3309 (elobixibat) includes details such as increases in spontaneous bowel movements (SBMs), reduced time to first SBM, and favorable adverse event profiles. While this is useful, consider:

  • Clarifying the context (e.g., patient populations studied in chronic idiopathic constipation vs. those with PBC/cholestatic itch).
  • Summarizing the clinical trial endpoints and statistical significance in a table or bullet format.
  • When mentioning adverse events (e.g., “mild abdominal pain in 19% of patients” and “diarrhea in 13%”), it may be helpful to indicate whether these events were dose-dependent or consistent across trials..”

Thank you for your invaluable comments. We agree with you. According to your suggestion, we revised Table 1 and our manuscript.

Reply to your comment 14: “The manuscript discusses how IBAT inhibitors reduce bile acid pool size and affect cholesterol homeostasis. This section could be enhanced by linking these mechanistic insights to the observed improvements in pruritus.”

Thank you for your invaluable comments. We agree with you. According to your suggestion, we revised our manuscript.

Reply to your comment 15: “The discussion of the effects on the fecal microbiome (e.g., changes in Firmicutes and Bacteroidetes) is intriguing. Consider expanding on how these microbiome shifts might contribute to therapeutic outcomes or potential side effects.”

Thank you for your invaluable comments. We agree with you. According to your suggestion, we revised our manuscript.

Reply to your comment 16: “16. The discussion on IBAT inhibitors in pediatric cholestatic conditions (e.g., Alagille syndrome and progressive familial intrahepatic cholestasis) is well detailed. However, the narrative would benefit from a clearer transition that distinguishes their use in pediatric populations versus adult PBC patients.”

Thank you for your invaluable comments. We agree with you. According to your suggestion, we revised Table 1 and our manuscript.

Reply to your comment 17: “Ensure consistency when referencing clinical studies (e.g., ICONIC study for maralixibat and phase 3 data for odevixibat). A brief summary of study designs (randomized, placebo-controlled, etc.) would help readers gauge the quality of evidence.”

Thank you for your invaluable comments. We agree with you. According to your suggestion, we revised our manuscript.

Reply to your comment 18: “The discussion on cholestyramine is generally clear, but note that there is a minor error in phrasing (“despite of the alleviating the itch sensation”). This should be rephrased (e.g., “Despite its ability to alleviate itch in some patients, complete resolution of chronic itch is rare.”).”

Thank you for your invaluable comments. We agree with you. According to your suggestion, we revised our manuscript.

Reply to your comment 19: “19. The mechanism of rifampin in reducing autotaxin activity is mentioned as “largely unknown yet.” Consider briefly discussing any hypotheses or known pharmacodynamic effects, even if they are preliminary.”

Thank you for your invaluable comments. We agree with you. According to your suggestion, we revised our manuscript.

Round 2

Reviewer 1 Report

Comments and Suggestions for Authors

As recommended in the first round of your manuscript, please focus your review only on cholestatic chronic liver diseases in different ages (childhood, pregnancy, and adults). Therefore, the section 4.1.2 IBAT inhibitor for chronic constipation is not the subject of this review.

Please for a better understanding and simplifying your subject

  1. Eliminate the section 4.1.2, the references 77-80, and shorten table 1.
  2. Include a new paragraph regarding Intrahepatic cholestasis of pregnancy

Reviewer 2 Report

Comments and Suggestions for Authors

The manuscript has been much improved after thorough revision. 

Reviewer 3 Report

Comments and Suggestions for Authors

The authors appropriately addressed all the issues I raised

Round 3

Reviewer 1 Report

Comments and Suggestions for Authors

The authors have made modifications to the original manuscript based on the reviewers’ comments and advice improving the quality of their study.

Now, the manuscript is suitable for publication in International Journal of Molecular Sciences